# Does Bicanalicular Intubation Improve the Outcome of Endoscopic Dacryocystorhinostomy?

**DOI:** 10.3390/jcm11185387

**Published:** 2022-09-14

**Authors:** Petr Matoušek, Jakub Lubojacký, Michaela Masárová, Lenka Čábalová, Stanislav Červenka, Pavel Komínek

**Affiliations:** 1Department of Otorhinolaryngology—Head and Neck Surgery, University Hospital Ostrava, 708 52 Ostrava, Czech Republic; 2Department of Craniofacial Surgery, Faculty of Medicine, University of Ostrava, 701 03 Ostrava, Czech Republic

**Keywords:** distal nasolacrimal duct obstruction, endoscopic dacryocystorhinostomy, bicanalicular intubation

## Abstract

The aim of this study was to compare the success of endoscopic dacryocystorinostomy with and without bicanalicular intubation in the treatment of distal nasolacrimal duct obstruction. Methods: In a prospective, randomized, comparative study, endoscopic dacryocystorinostomy without bicanalicular silicone intubation (Group I) and endoscopic dacryocystorinostomy with intubation (Group II) were performed in patients with distal nasolacrimal duct obstructions. The tubes were removed 3 months after surgery in Group II, and the patients were followed up for 6 months after surgery. Therapeutic success was defined as the fluorescein dye disappearance test grade 0–1 corresponding with a complete resolution of symptoms. Results: Thirty patients, aged 23–86 years, were included in the study. The success rate was 13/15 (86.67%) in Group I and in 14/15 (93.33%) in Group II. The difference between the two groups was not statistically significant (*p* = 0.483). Most common complications were granulations that occurred in 1/15 (6.67%) patient in Group I and in 2/15 (13.33%) patients in Group II. Adhesions in rhinostomy with epiphora and persistent secretion were observed in 1/15 (6.67%) patient in Group II. Conclusions: Bicanalicular intubation does not significantly increase the success rate of EDCR in distal nasolacrimal obstruction in adults.

## 1. Introduction

Endoscopic dacryocystorhinostomy (EDCR) and external dacryocystorinostomy is currently the treatment of choice in distal obstruction of the lacrimal duct [1,2,3].

EDCR has a high success rate, ranging from 85–99% depending on the technique, site of obstruction, and use of a silicone tube [3]. The most common reasons for treatment failure are tissue granulation and adhesions at the site of the rhinostomy, and formation of obstruction in the common canaliculus [1]. Bicanalicular intubation with silicone tube has been described by several authors in efforts to improve EDCR outcomes and ensure stoma patency after surgery [1,2,4].

Although silicone tubes are widely recommended for revised surgery in pediatrics, their benefit is disputed in adult patients [4,5,6,7,8,9,10,11]. This work aims to evaluate and compare the success of EDCR with and without bicanalicular intubation in the treatment of distal nasolacrimal duct obstruction.

## 2. Materials and Methods

A prospective randomized study was conducted at University Hospital Ostrava from January 2018 to January 2021. Patients were randomly assigned by coin flip randomization to 2 groups. Group I comprised 15 patients without intubation; Group II comprised 15 patients with bicanalicular intubation.

The study included patients over 18 years of age with distal obstruction in the nasolacrimal duct and an absence of pathology at the punctums, upper and lower canaliculi and common canaliculus. The patients under 18 years old, with previous nasolacrimal surgery, nasal or paranasal sinus trauma or malignancy, and resulting systemic disease were excluded from the study.

After taking history, each patient underwent detailed ENT examination including nasal endoscopy. Lacrimal duct obstruction was confirmed with the fluorescein dye disappearance test (FDDT) [12]. Probing and lacrimal irrigation of the lower canaliculus determined the site of the obstruction [13].

EDCR was performed under general anesthesia. After vasoconstriction of nasal mucosa with adrenaline 1:100,000 the position of the lacrimal sac was marked with bayonet forceps. The mucous membrane above the frontal process of maxilla in front of the head of the middle turbinate is incised, elevated and removed with Blackesley forceps and the bone medial is exposed to the lacrimal sac. The bone laying medial and anterior to the sac is removed using Kerrison rongeurs to widely expose the lacrimal sac to the nasal cavity.

After the lacrimal sac had been exposed, the Bowman probe is placed through the canaliculus into the sac and the moving of the sac can then usually be well viewed intranasally. The sac is incised and opened with a sickle knife and medial wall of the sac is removed [7].

Bicanalicular intubation was carried out in Group II patients: a silicone tube (FCI, Paris, France) was placed into lacrimal system through both canaliculi. The ends were tied together with 3 knots in the nasal cavity. No packing was placed after surgery.

Patients carried out nasal douching with warm saline for 6–8 weeks after surgery. No additional therapy (local antibiotics, corticoids) was used in a postoperative period. The tube in Group II patients was removed 3 months after surgery. Patients were followed up 1 week, 1 month, 3 months, and 6 months after surgery. Functional success was defined as a normal FDDT and the absence of epiphora and dacryocystitis 6 months after surgery. Failure was defined as the absence of improvement or the worsening of the symptoms.

Complications as granulation, adhesions in the rhinostomy or in the puncta, canalicular lacerations or tube dislodgement were observed.

For the statistical analysis, the Mann–Whitney U-test for quantitative data and Fisher’s exact test for the frequency of categorical data were used.

## 3. Results

Thirty patients, 9 men and 21 women, with a mean age of 52 years, were studied. EDCR without intubation (Group I) was performed in 15 patients and EDCR with bicanalicular intubation (Group II) was performed in 15 patients (Table 1). There was no statical difference between groups in terms of age, side or sex (*p* > 0.05).

Functional success was observed in 13/15 (86.67%) patients in Group I and in 14/15 (93.33%) patients in Group II, the difference between both groups was not statistically significant (*p* = 0.483). Postoperatively, all patients from both study groups remained free of dacryocystitis. No dislodgement of the tube, canalicular lacerations or adhesions in the puncta were observed.

Six months after, EDCR revision surgery was required in 1/15 (6.67%) of patients from Group I due to granulation occluding the rhinostomy resulting in recurrent epiphora. Granulations in patients from Group II occurred in 2/15 (13.3%) of cases, no intervention was needed since the granulations did not occlude the rhinostomy.

Three months after surgery, rhinostomy adhesion resulting in epiphora and persistent mucopurulent secretion in the lacrimal sac was observed in 1/15 (6.67%) of patients from Group II. It was resolved with endonasal incision of the lacrimal sac and the concomitant removal of the tube. At further postoperative follow-up, the ostium was open and the patient subjectively free of clinical complaints. There was no statistical difference between the two groups.

## 4. Discussion

In recent years, endonasal dacryocystorhinostomy has gained popularity among surgeons in the treatment of subsaccal obstruction of the nasolacrimal duct thanks to its many advantages, including its high efficacy, low risk of complications, and shorter hospital stay [1].

The most common causes of EDCR failure are granulomatous formations and adhesions at the site of the rhinostomy, and obstruction in the common canaliculi [1,2,4]. EDCR with bicanalicular intubation with a silicone tube is frequently employed to ensure patency of the rhinostomy, although opinion is divided over whether this approach brings any real benefit [1,2,4]. It has been claimed that bicanalicular intubation can improve treatment outcomes, while others argue that silicone stenting can complicate postoperative recovery by predisposing the site to tissue granulation, infection, canalicular lacerations, or adhesions in the puncta, leading to treatment failure [1,2,3,4,5].

In efforts to identify the most appropriate surgical method, several studies comparing EDCR with and without bicanalicular intubation have been published in recent years (Table 2). In many cases the results of these returned no differences of statistical significance, success rates being comparably high for both treatment modalities [8,9,10,11,13].

In a prospective study of 118 patients from 2013, Chong et al. present a 96.3% success rate in the intubated group; the success rate was almost as high (95.3%) in the nonintubated group [8]. Very high success rates have been described by other authors. A study of 173 cases by Al-Qahtani et al. achieved treatment success in 96% of intubated patients and 91% of nonintubated patients [2]. In a study of 50 patients in 2008, Hagurop et al. achieved treatment success in 96% of the intubated group and 93.3% in the group without intubation [8]. Slightly worse results were achieved by Yeon et al. in 2012 with a smaller study group of 36 patients, although even in this study there was no significant difference in treatment outcome at 84.2% with intubation vs. 81.8% without [10]. The contention that bicanalicular intubation does not influence EDCR outcome in subsaccal stenosis is supported by our own experience as well as that of other authors [11,13].

Other authors have observed certain, although insignificant, differences between study groups with and without intubation [14,15,16,17,18,19]. While some authors recorded better outcomes in patients without intubation, others have found EDCR to be more successful in combination with bicanalicular intubation [14,15,16,17,18,19].

**Table 2 jcm-11-05387-t002:** Characteristics of studies, comparison of the success rate between endoscopic dacryocystorinostomy with and without silicone intubation.

Author	Year	No of Pts (Eyes)	Succes Rate with Tube	Succes Rate without Tube	*p* Value	Tube Removed *	Follow-Up *
Smirnov et al. [13]	2008	46	18/23 (78.2%)	23/23 (100%)	*p* < 0.049	2	6
Harugop et al. [9]	2008	290	48/50 (96%)	224/240 (93.3%)	*p* = 0.4317	3	16
Unlu et al. [11]	2009	38	16/19 (84.2%)	18/19 (94.7)	*p* = 0.123	2	100
Al-Qahtani et al. [2]	2012	173	89/92 (96.7%)	73/81 (90.1%)	*p* = 0.117	4	12
Yeon et al. [10]	2012	36 (41)	5/6 (83%)	14/15 (93.3%)	*p* = 0.50	1	9
Chong et al. [8]	2013	118 (128)	61/63 (96.3%)	62/65 (95.3%)	*p* = 0.79	2	12
Shashidhar et al. [18]	2014	57 (62)	30/32 (93.7%)	26/30 (86.6%)	*p* = 0.4810	1.5	6
Reddy et al. [17]	2015	20	9/10 (90%)	8/10 (80%)		1.5	6
Longari et al. [14]	2016	99	45 (82.2%)	44 (88.6%)	*p* > 0.05	2	18
Rao et al. [16]	2016	50	23/25 (92%)	21/25 (84%)	*p* > 0.05	3	6
Fayers et al. [15]	2016	300	144/152 (94.7%)	130/148 (87.8%)	*p* = 0.034	3	12
Matoušek et al. (this paper)	2022	30	13/15 (86.6%)	14/15 (93.3%)	*p* = 0.483	3	6

* months.

In 2008, Smirnov et al. published a prospective randomized study in which 42 patients were randomized into two study groups. Results were evaluated 6 months after surgery and the treatment success was 100% without intubation and just 78% with intubation [13]. In a 4-year retrospective study of 84 patients (a total of 89 operations) published in 2016, Longari et al. found that EDCR success rates were lower with intubation (82.2%) than without intubation (88.6%) [15].

Conversely, a large-scale study by Fayers et al. in 2016 involving 300 patients noted treatment success in 94.7% of intubated patients and 87.8% of nonintubated patients [15]. In the same year, similar results were obtained by Rao et al. in a study of 50 patients. A higher success rate (92 vs. 84%) was observed in the intubated group [17]. Likewise, other studies, although with smaller sample sizes (Reddy et al. and Shashidhar et al.) have concluded that EDCR with nasolacrimal duct intubation is more efficacious [18,19].

In the results, it is possible to see a certain discrepancy between individual authors [14,15,16,17,18,19]. The findings presented above are summarized in a comprehensive meta-analysis of 12 randomized controlled studies from 2007–2016 which was published by Kang et al. It was determined that the success rate of EDCR with bicanalicular intubation improved from 2012 onwards, but with no significant differences between intubated and nonintubated study groups in terms of efficacy or incidence of complications [4].

Across the literature, no significant differences have yet been established between groups with and without intubation; only in isolated cases have differences of 10–20% been found, and only with smaller sample sizes. It can therefore be stated that, according to the available literature, the benefits of intubation alongside EDCR for distal stenosis are at best marginal. The impact of other post-surgical therapies is also studied, such as local or systemic antibiotics, steroids, nasal decongestants. The results demonstrate no meaningful influence, but should be considered critically [20].

According to our initial results, treatment was successful in 93.33% in Group II and 86.67% in Group I, but the difference was not statistically significant (*p* = 0.483) due to small sample size. In Group I, reoperation was necessary in one patient due to granulation of the rhinostomy. The question of whether intubation could have prevented this complication thus arises. On the other hand, the silicone stent was most likely the cause of recurrent epiphora, and mucopurulent accumulation in the lacrimal sac 3 months after surgery in a Group II patient. We conclude that bicanalicular intubation is not indicated in EDCR for distal obstruction and should be performed only in selected cases with obstruction of the common canaliculi.

Despite our initial findings being limited by the small sample size, the results are consistent with the wider literature. Further study comparing larger sample sizes is necessary.

## 5. Conclusions

This initial study demonstrates that bicanalicular intubation does not increase the success rate of EDCR in distal nasolacrimal obstruction. The risk of postoperative complications is not affected by the use of silicone stenting.

## Figures and Tables

**Table 1 jcm-11-05387-t001:** Results of endoscopic dacryocystorinostomy with and without bicanalicular intubation (follow-up 6 months). The difference between the two groups was not statistically significant (*p* = 0.483).

Outcome	Group I EDCR without Tube (15 Cases)	Group II EDCR with Tube (15 Cases)
Success	13 (86.67%)	14 (93.33%)
Failure	2 (13.33%)	1 (6.67%)

## Data Availability

Not applicable.

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
