# Peer review of "Does Bicanalicular Intubation Improve the Outcome of Endoscopic Dacryocystorhinostomy?"

_jcm, 2022, doi:10.3390/jcm11185387_

Round 1
Reviewer 1 Report
Dear authors,
it is a well-conducted and well-written study investigating the possible benefit of intraoperative intubation of the lacrimal ducts. However, the case number calculation is missing, so that the missing proof of a significant difference in the results can be caused by the low number of cases. Discussion seems to be to long, as the table summarizing the results of other studies on this issue, gives us sufficient information.
The biggest disadvantage of the study is the lack of its novelty.
Kind regards
Author Response
Response to reviewer 1
It is a well-conducted and well-written study investigating the possible benefit of intraoperative intubation of the lacrimal ducts. However, the case number calculation is missing, so that the missing proof of a significant difference in the results can be caused by the low number of cases. Discussion seems to be to long, as the table summarizing the results of other studies on this issue, gives us sufficient information. Corrected, added statistical significancy and p value and added descriptive numbers.
Reviewer 2 Report
In this prospective work, the authors demonstrated that the bicanalicular silicone intubation is not essential in the final surgical outcome (both functional and anatomical) of END-DCR.
This topic has been largely analyzed; however, the prospective nature add value to the work. Nevertheless, major topics should be addressed before any further consideration.
- What is subsaccal obstruction? I suggest the authors to use more conventional terms such as distal or post-canalicular lacrimal obstruction.
- I encourage the authors to use more updated literature citation, none of them are dated more than 2018. For example, line 1 and 2, the authors stated that END-DCR is currently the treatment of choice for distal obstructions. This is not true, END-DCR can be equally compared to EXT-DCR. So please re-phrase and add the most recent meta-analysis on this topic (Vinciguerra A et al. Best treatments available for distal acquired lacrimal obstruction: A systematic review and meta-analysis. Clin Otolaryngol. 2020 Jul;45(4):545-557).
Methods:
- How was randomization performed? This is an important information that is lacking.
- I don’t understand why the authors used a non parametric test and a Chi-square test; I would rather suggest to use the Mann-Whitney and a fisher exact (eventually associated with a cramer’v value), considering the low number of patients enrolled in the study.
Results:
- Why there are not the p-value of the statistical analysis? It is of paramount importance to show the most relevant data in such section.
Discussion
- Table 2: the label stated comparison of success rate, but it is hardly the case that all papers analyzed both the functional and anatomical success, that are two completely different outcomes. Most of the time, the anatomical outcome is considered. So please modify the table accordingly to this fundamental aspect. In addition, I find hard to interpretate the results of each study. I would rather suggest to add the p-value of each study.
- Add p-values of your result in the discussion as well
- Among the discussion I suggest to add a paragraph concerning the post-operative therapies in general (ABT, corticosteroid, MMC etc…), you could use another recent meta-analysis on this topic (Vinciguerra A, et al. Impact of Post-Surgical Therapies on Endoscopic and External Dacryocystorhinostomy: Systematic Review and Meta-Analysis. Am J Rhinol Allergy. 2020 Nov;34(6):846-856.)
Level of English not sufficient, please revise it.
Author Response
Comments to review:
- What is subsaccal obstruction? I suggest the authors to use more conventional terms such as distal or post-canalicular lacrimal obstruction. Corrected – distal lacrimal obstruction.
- I encourage the authors to use more updated literature citation, none of them are dated more than 2018. For example, line 1 and 2, the authors stated that END-DCR is currently the treatment of choice for distal obstructions. This is not true, END-DCR can be equally compared to EXT-DCR. So please re-phrase and add the most recent meta-analysis on this topic (Vinciguerra A et al. Best treatments available for distal acquired lacrimal obstruction: A systematic review and meta-analysis. Clin Otolaryngol. 2020 Jul;45(4):545-557).
Corrected - we updated literature and changed statement for both ext- and en-DCR, we added most recent meta analysis.
Methods:
- How was randomization performed? This is an important information that is lacking. Randomization was performed by coin flip - information is added.
- I don’t understand why the authors used a non parametric test and a Chi-square test; I would rather suggest to use the Mann-Whitney and a fisher exact (eventually associated with a cramer’v value), considering the low number of patients enrolled in the study. Statistical analysis was performed by statistician - we changed test according your reccomendation. The results are the same for both tests.
Results:
- Why there are not the p-value of the statistical analysis? It is of paramount importance to show the most relevant data in such section. Corrected. We added information - p value
Discussion
- Table 2: the label stated comparison of success rate, but it is hardly the case that all papers analyzed both the functional and anatomical success, that are two completely different outcomes. Most of the time, the anatomical outcome is considered. So please modify the table accordingly to this fundamental aspect. In addition, I find hard to interpretate the results of each study. I would rather suggest to add the p-value of each study. Corrected.
- Add p-values of your result in the discussion as well - added.
- Among the discussion I suggest to add a paragraph concerning the post-operative therapies in general (ABT, corticosteroid, MMC etc…), you could use another recent meta-analysis on this topic (Vinciguerra A, et al. Impact of Post-Surgical Therapies on Endoscopic and External Dacryocystorhinostomy: Systematic Review and Meta-Analysis. Am J Rhinol Allergy. 2020 Nov;34(6):846-856.) - We added paragraph about post surgical therapy and added citation to literature.
Level of English not sufficient, please revise it. The article was checked by native speaker.
Round 2
Reviewer 2 Report
The paper has been reviewed properly and it is now worthy fo publication.